# Sublimed $C_{60}$ for efficient and repeatable perovskite-based solar cells

Ahmed A. Said [1,6] ✉, Erkan Aydin [1,6] ✉, Esma Ugur [1], Zhaojian Xu [2], Caner Deger [3], Badri Vishal[1], Aleš Vlk[4], Pia Dally[1], Bumin K. Yildirim [1], Randi Azmi [1], Jiang Liu[1], Edward A. Jackson[5], Holly M. Johnson[2], Manting Gui[2], Henning Richter[5], Anil R. Pininti[1], Helen Bristow[1], Maxime Babics [1], Arsalan Razzaq[1], Suman Mandal[1], Thomas G. Allen[1], Thomas D. Anthopoulos[1], Martin Ledinský[4], Ilhan Yavuz [3], Barry P. Rand [2] & Stefaan De Wolf [1] ✉

Thermally evaporated $C_{60}$ is a near-ubiquitous electron transport layer in state-of-the-art $p–i–n$ perovskite-based solar cells. As perovskite photovoltaic technologies are moving toward industrialization, batch-to-batch reproducibility of device performances becomes crucial. Here, we show that commercial as-received (99.75% pure) $C_{60}$ source materials may coalesce during repeated thermal evaporation processes, jeopardizing such reproducibility. We find that the coalescence is due to oxygen present in the initial source powder and leads to the formation of deep states within the perovskite bandgap, resulting in a systematic decrease in solar cell performance. However, further purification (through sublimation) of the $C_{60}$ to 99.95% before evaporation is found to hinder coalescence, with the associated solar cell performances being fully reproducible after repeated processing. We verify the universality of this behavior on perovskite/silicon tandem solar cells by demonstrating their open-circuit voltages and fill factors to remain at 1950 mV and 81% respectively, over eight repeated processes using the same sublimed $C_{60}$ source material. Notably, one of these cells achieved a certified power conversion efficiency of 30.9%. These findings provide insights crucial for the advancement of perovskite photovoltaic technologies towards scaled production with high process yield.

Perovskite-based solar cells (PSCs) are emerging high-efficiency photovoltaic (PV) technologies on the verge of commercialization[1,2]. In their single-junction (1-J) implementation, initial PSCs were fabricated in the so-called $n\text{-}i\text{-}p$ architecture, i.e. by first depositing the electron transport layer (ETL, $n$-layer), followed by the perovskite absorber ($i$-layer), and hole transport layer (HTL, $p$-layer). So far, majority of the certified champion 1-J cells have been reported in this configuration[3,4]. However, in recent years, $p\text{-}i\text{-}n$ devices have drawn more attention due to their compatibility with low-temperature processing, enabling applications such as tandem solar cells[5–8], and more promising material choices for the HTL in terms of device stability and materials cost[9]. In this configuration, $C_{60}$-based

[1]King Abdullah University of Science and Technology (KAUST), KAUST Solar Center (KSC), Physical Science and Engineering Division (PSE), Thuwal 23955-6900, Kingdom of Saudi Arabia. [2]Department of Electrical and Computer Engineering, Princeton University, Princeton, NJ 08544, USA. [3]Department of Physics, Marmara University, Istanbul, Türkiye. [4]Laboratory of Nanostructures and Nanomaterials, Institute of Physics, Academy of Sciences of the Czech Republic, v. v. i., Cukrovarnická 10, Prague 162 00, Czech Republic. [5]Nano-C, Inc., 33 Southwest Park, Westwood, MA 02090, USA. [6]These authors contributed equally: Ahmed A. Said, Erkan Aydin. ✉e-mail: ahmedali.ahmed@kaust.edu.sa; erkan.aydin@kaust.edu.sa; stefaan.dewolf@kaust.edu.sa

fullerenes are usually chosen as the ETL[8,10–19], although several alternative materials have been explored as well[20–28].

The preference of $C_{60}$ as the ETL for *p-i-n* PSCs largely relates to their small conduction band offset and large valence band offset with respect to the perovskite, resulting in favorable electron-extraction and hole-blocking properties, respectively, essential for a good electron-selective contact. Other advantages are their high electron mobility, hydrophobic nature, and capability to passivate perovskite surface antisite defects[10–13]. Their poor solubility is usually resolved by synthesizing $C_{60}$ to feature specific functional groups[29]. This enables processing with solution-based methods such as spin-coating, spray, or blade coating[30,31]. Thermal evaporation of $C_{60}$ is possible too and is arguably the industrially-preferred method for perovskite device fabrication as this technique enables good conformality and thickness control on rough and topologically complex surfaces. This is of specific importance for multijunction perovskite/silicon applications in the common *p-i-n* architecture, where accurate thickness and conformality control are required to minimize parasitic optical absorption caused by $C_{60}$ while maintaining its essential electronic role at the ETL in the device[32].

In this work, we investigate the evolution of the electronic quality of $C_{60}$ thin films subjected to repeated evaporation processes and gauge the impact of such a procedure on device performance. Our findings reveal that, as a result of the heating and cooling cycles of the source material during multiple evaporation cycles, $C_{60}$ undergoes a conversion into higher molecular weight structures through the fusion of $C_{60}$ molecules. This transformation leads to modifications in the electronic properties of the fullerene, detrimentally affecting device performance. However, we show that further purification of

as-received $C_{60}$ (through a sublimation process, before the evaporation) can help to avoid these issues, and device performance remains unaffected even after repeated deposition cycles. The insights obtained from this research are crucial for the advancement towards practical, scaled applications of perovskite-based *p-i-n* structured single-junction and tandem PV technologies that employ evaporated $C_{60}$ as the ETL.

## Results

### Device performance with repeated processes of as-received $C_{60}$

We first assessed the performance of commercial as-received $C_{60}$ source material (henceforth referred to as "as-received") with repeated deposition cycles directly on 1-J PSCs. For this, we used a device stack in the ITO/NiO$_x$/MeO-2PACz/perovskite/$C_{60}$/BCP/Ag configuration, as sketched in Fig. 1a, where ITO is indium tin oxide, NiO$_x$ is nickel oxide, MeO-2PACz is (2-(3,6-dimethoxy-9H-carbazol-9-yl)ethyl)phosphonic acid and BCP is bathocuproine. One thermal cycle involves evacuating the evaporation chamber (to a base pressure of $10^{-7}$ Torr), followed by thermal evaporation of $C_{60}$ layers (crucible-temperature, $T_C$, and time-temperature profiles are given in Supplementary Fig. 1a), and finished by bringing the half-finished device (NiO$_x$/MeO-2PACz/perovskite/$C_{60}$) to the load lock pressure (around $10^{-3}$ Torr). We observed that the maximum $T_C$ during the deposition needed to sustain a constant deposition rate increased with every new cycle. We plotted the rise of the set temperature to achieve the 0.1 Å/s (and then 0.25 Å/s) deposition rate in Supplementary Fig. 1a, which is the standard baseline $C_{60}$ evaporation process in our lab. After the 8th cycle, we also found that the color of the powder in the crucible changed from black to brown as shown in Supplementary Fig. 1b. From the current density-voltage ($J-V$) analysis of the devices, we found an average 5 mV drop in $V_{oc}$ for

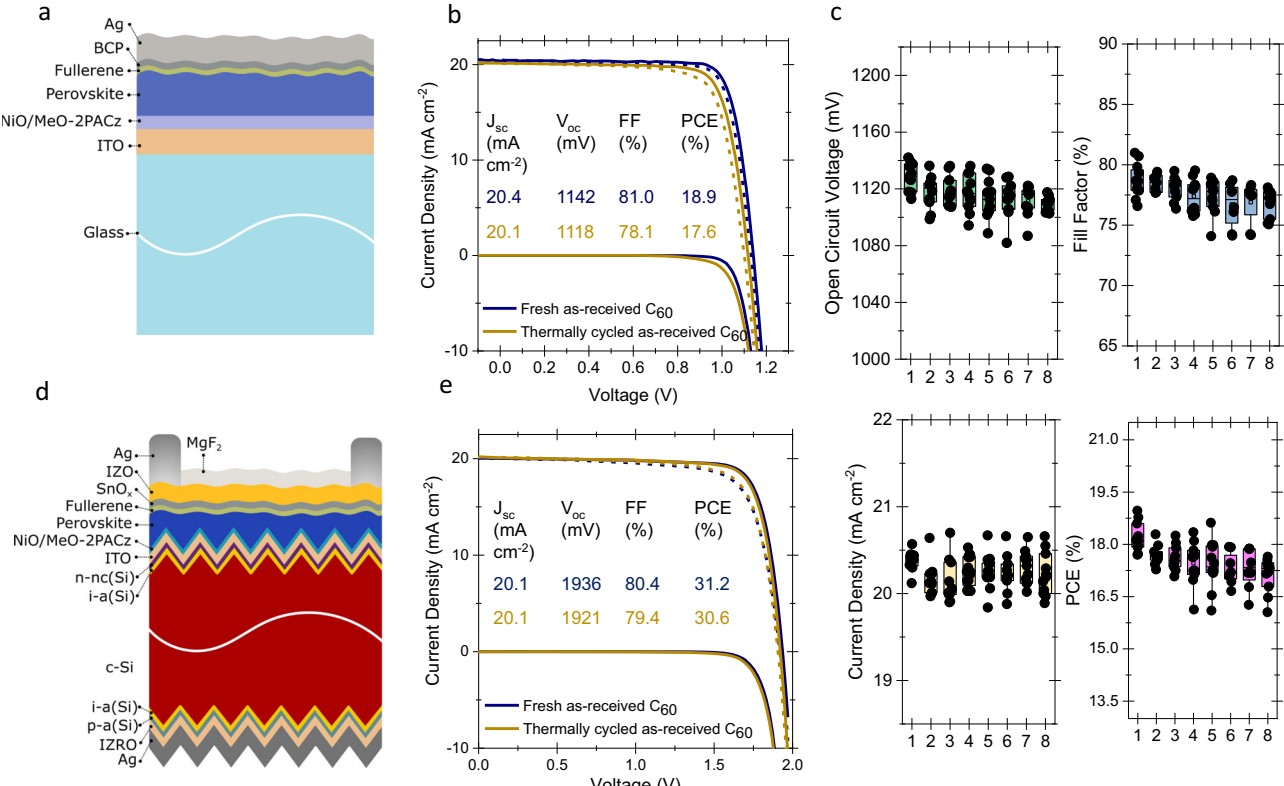

**Fig. 1 | Perovskite solar cells fabricated with commercial as-received $C_{60}$.**
**a** Device structure of the tested single-junction perovskite solar cells, **b** corresponding *J-V* curves, and **c** statistical distributions of the photovoltaic characteristics. The x-axis value represents the deposition cycle. Here, we grouped co-deposited, half-finished devices (ITO/NiO$_x$/MeO-2PACz/perovskite), to be finished with $C_{60}$ layers using the same powder but subsequently evaporated.

**d** A sketch of perovskite/silicon tandem solar cells, and **e** their corresponding device characteristics with $C_{60}$ contacts with fresh and thermally cycled powders. Here, the perovskite layers of single-junction PSCs were fabricated using a hybrid method[52] that consists of a two-step process for the perovskite formation. Initially, an inorganic template was evaporated, and then a solution conversion step was employed to accomplish conversion into the perovskite phase.

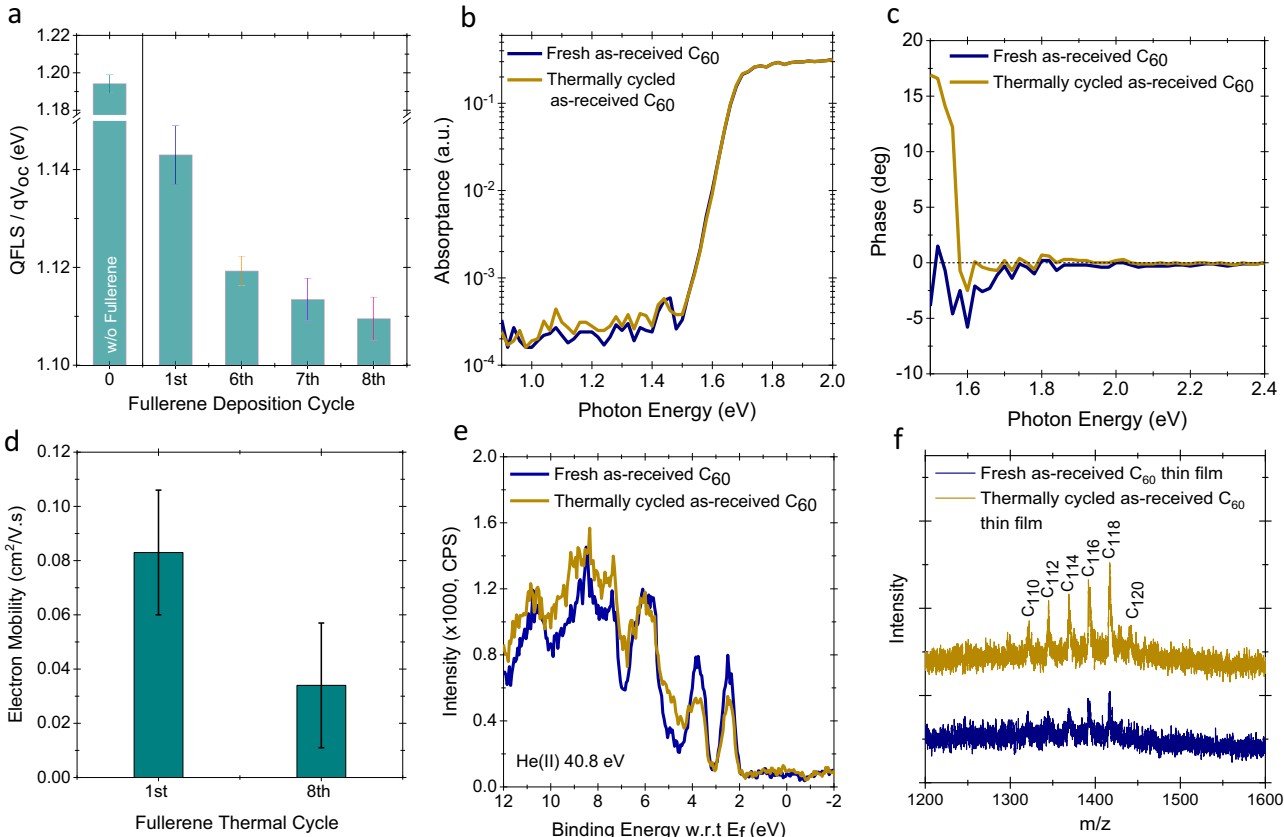

**Fig. 2 | Optoelectronic properties of commercial as-received $C_{60}$ films. a** QFLS values for the stack of ITO/NiO$_x$/MeO-2PACz/perovskite/$C_{60}$ at various $C_{60}$ deposition cycles, together with without $C_{60}$ stack. **b** PDS absorption spectra of the quartz/perovskite/ $C_{60}$ stacks, and **c** its phase shift compared to excitation. **d** Electron mobility values of the $C_{60}$ layers at 1$^{st}$ and 8$^{th}$ cycles measured via FETs.

**e** UPS spectra of the $C_{60}$ thin films on c-Si wafer showing the distribution of the energy states close to the HOMO level. **f** MALDI-TOF analysis on thin films produced from both fresh and thermally cycled powders. To perform this analysis, the films were deposited on glass substrates and subsequently scratched to collect the resulting powder, which was then utilized for the MALDI-TOF analysis.

the corresponding devices after each thermal cycle, as well as a gradual decrease in *FF*. The $V_{oc}$ of the device in the 1$^{st}$ thermal cycle was 1142 mV (average is 1130 mV), which dropped to 1118 mV (average is 1109 mV) after the 8$^{th}$ thermal cycle, as shown in Fig. 1b and c. We confirmed this trend of declining $V_{oc}$ and *FF* with increasing $C_{60}$ deposition cycles for other types of PSCs, for instance, those based on one-step solution processed Cs$_{0.05}$FA$_{0.8}$MA$_{0.15}$Pb(I$_{0.745}$Br$_{0.255}$)$_3$ (1.68 eV) and Cs$_{0.03}$(FA$_{0.90}$MA$_{0.10}$)$_{0.97}$PbI$_3$ (1.55 eV) perovskite absorbers (Supplementary Fig. 2a and 2b, respectively)[33,34]. Then, we repeated the same experiment with 1.68 eV bandgap hybrid processed perovskite as commonly used in perovskite/silicon tandem solar cells (see Methods for the details), and found the same trend for $V_{oc}$ and *FF* performance drops as shown in Supplementary Fig. 2c.

We also verified this behavior for larger device areas (0.1 cm$^2$–1 cm$^2$) on perovskite/silicon tandem solar cells with the structure given in Fig. 1d. Again, we observed $V_{oc}$ and *FF* losses after repeated processing, as depicted in the statistical analysis of the device parameters in Supplementary Fig. 3. The characteristic *J-V* curves of the tandem cells and their external quantum efficiency (EQE) spectra are shown in Fig. 1e and Supplementary Fig. 4, respectively. The tandems show a similar performance drop to the 1-J perovskite devices after repeated thermal cycles of the $C_{60}$ ETL. The quasi-Fermi level splitting (QFLS) from the half-finished device stacks (NiO$_x$/MeO-2PACz/perovskite/$C_{60}$) after the 1$^{st}$, 6$^{th}$, 7$^{th}$, and 8$^{th}$ deposition cycles confirmed the voltage losses on devices stacks, as we again observed ~5 meV loss after each cycle (Fig. 2a).

The origin of the observed voltage losses after thermal cycling is likely due to a change in the structural and electronic properties of

the $C_{60}$ layer itself and the interface it forms with the perovskite. To gain further insight, we first characterized quartz/perovskite/$C_{60}$ stacks via photothermal deflection spectroscopy (PDS) to probe whether any deep states were formed or induced in the perovskite band-gap upon contact with the fullerene. We found that the perovskite band edge is almost identical for fresh and thermally cycled samples, indicating that the $C_{60}$ films do not alter the bulk properties of the perovskite, as expected. The absorption level within the bandgap (from 1 to 1.4 eV) as shown in Fig. 2b is also very similar. We then analyzed the phase changes between the pump beam intensity and the detected signal in PDS measurements, which is a measure of the time shift between the light excitation and the measured PDS signal. As depicted in Fig. 2c, we set the time shift to 0° at the high absorption region (>2.2 eV), where all the incoming light is absorbed by both samples near the perovskite surface. At the band edge area, the phase of fresh $C_{60}$ samples decreases, which suggests that the light absorption depth increases and therefore excited phonons need more time to travel to the sample surface. This behavior is standard for most PDS measurements. However, in the case of the thermally cycled sample, the phase decrease is minimal and is followed by a strong increase in the phase shift. This means that the thermal signal generation is concentrated at the very top of the sample, close to the surface. This may be caused either by direct absorption of the light at the surface states or by recombination of electron-hole pairs generated within the perovskite film at the surface states. From here, we can infer that fullerene contacts deposited with thermal cycles lead to a higher concentration of electronically active defect states at the perovskite/fullerene interface.

Next, we measured the electron mobility of the $C_{60}$ layers deposited during both the first and last thermal cycles via top-gate bottom-contact field effect transistors (FETs). The architecture of the fabricated FET is shown in Supplementary Fig. 5. The electron mobilities ($\mu_e$) of the 1st deposition (0.083 $cm^2$/V s) are higher than the 8th deposition (0.034 $cm^2$/V s), on average, as shown in Fig. 2d. This implies that the structure of the $C_{60}$ layers might have changed during repeated thermal cycles. The work function (WF) and the highest occupied molecular orbital (HOMO) of the distinct films are also other parameters to be considered, which we investigated via ultraviolet photoelectron spectroscopy (UPS) measurements. We obtained a WF value of -4.4 eV for both the fresh and the thermally cycled $C_{60}$ films. The valence band maximum, determined with a Gaussian fitting method, also remains unchanged (6.4 eV), as shown in (Supplementary Fig. 6a) and Fig. 2e, respectively. However, we observed a change in the density of states (DOS) on the thermally cycled samples as shown in Fig. 2e. From these characterizations, we conclude that the properties of the $C_{60}$ powder, and therefore the deposited thin films, have changed with thermal cycling. This is mainly manifested through additional recombination states, in line with the mobility results of the FETs.

Subsequently, we collected the brownish powder from the crucible after the 8th cycle for further analysis. We first transferred those powders onto quartz glass substrates, sandwiched them with another glass sheet, and verified the PL emission via a hyperspectral imaging system. We observed that the thermally cycled powders showed an emission peak at 835 nm in addition to the main peak typical for pure $C_{60}$ located at 742 nm, as shown in Supplementary Fig. 6b[35,36]. The increased oxygen content for the thermally cycled samples, obtained from X-ray photoelectron spectroscopy (XPS) analysis, reveals the possibility of a chemical reaction occurring during the repeated evaporation processes (Supplementary Fig. 6c, d).

Further analysis indicates that the thermally cycled $C_{60}$ powder has a wider full width at half maximum (FWHM) of its main peak in the X-ray diffraction (XRD) spectrum. Moreover, a new feature at 11.4 degrees is present, as shown in Supplementary Fig. 7a, b, which is related to the formation of fullerene dimers and different fullerene derivatives at a certain fraction in the powder[37]. We also utilized the matrix-assisted laser desorption ionization-time of flight (MALDI-TOF) mass spectroscopy technique to investigate the change in molecular weights of the powders. Both fresh and thermally cycled $C_{60}$ powders showed a strong peak at 720 mass/charge number (m/z) which is assigned to the presence of a high fraction of $C_{60}$ in the sample bed. In addition, a weak peak for $C_{60}$ with $C^{13}$ at 721 m/z is observed in Supplementary Fig. 8. Interestingly, the thermally cycled $C_{60}$ powder showed additional peaks at 1444, 1465, 1489, 1513, and 1537 m/z. These peaks show the presence of additional fullerene derivatives such as $C_{120}$, $C_{122}$, $C_{124}$, $C_{126}$, and $C_{128}$, respectively with different numbers of the $C^{13}$ isotope[38,39], (Supplementary Fig. 8). The peak at 1444 m/z might be ascribed to fullerene with $C_{119}O$. After conducting MALDI-TOF analysis on the thin films, it became evident that films produced from the 1st deposition cycle of the $C_{60}$ powder lacked $C_{120}$ fragments. In contrast, the MALDI-TOF analysis of the 8th deposition cycle thin film showed a peak at 1441 m/z which is ascribed to $C_{120}$. We attribute this disparity to the alteration of evaporation temperatures. Furthermore, 1322, 1345, 1369, 1392, and 1417 m/z are found to be present in both $C_{60}$ thin films with different peak intensities and are assigned to $C_{110}$, $C_{112}$, $C_{114}$, $C_{116}$, and $C_{118}$, respectively. Importantly, the peaks of the thermally cycled thin film are stronger than that of fresh thin film, which confirms that the concentration of the higher molecular weight fullerenes in the thermally cycled thin film is higher than that of fresh thin film as shown in Fig. 2f. These findings explain the lower electron mobility of the thermally cycled $C_{60}$ thin film compared to that of the fresh thin film. The surface morphology of fresh and thermally cycled $C_{60}$ films was investigated by using atomic force microscopy (AFM). There is no

significant change in the morphology of fresh and thermally cycled $C_{60}$ films as shown in Supplementary Fig. 9a, b, respectively.

We performed further analysis of the thermally cycled powders with high-resolution transmission electron microscopy (HRTEM) as shown in Fig. 3a, Supplementary Fig. 10a and 10b at a low acceleration voltage (80 kV) as the $C_{60}$ molecules are electron-beam sensitive under conventional TEM conditions. HRTEM and Fast Fourier transform (FFT) spots matching with $C_{60}$ d-spacing ([111], [222], [602], [522], [511], [624], [800]) confirm that the $C_{60}$ is molecularly highly crystalline with a long-range spatial ordering (Fig. 3a and Supplementary Fig. 10c). From the HRTEM images given in Fig. 3b, we identified $C_{60}$, $C_{120}$, and $C_{60}$-dimers via the average diagonal diameter of the resolved molecules with the approach of Goel et al.[40].

Following this methodology, molecularly resolved HRTEM of $C_{60}$ and $C_{120}$ appeared with an average diameter of ~0.686 and 1.014 nm, respectively, with an accuracy of ± 0.038 nm. Also, a possible $C_{60}$-dimer with an intermolecular distance of 0.477 nm was observed as shown in Fig. 3c. The formation of high molecular weight fullerene derivatives in the presence of molecular oxygen has been explained by molecular dynamics simulations previously[41]. It is argued that molecular oxygen can react with the carbon-carbon bond in the pentagonal ring of low molecular weight fullerene, resulting in the formation of a carbonyl group bonded to fullerene. The carbonyl group then interconverts to an epoxide radical which attacks other fullerenes, forming higher molecular weight fullerene derivatives.

Next, we studied the correlation between the electronic properties and these structural changes via density functional theory (DFT) calculations. Here, for each case, we considered the thermodynamically most favorable high-symmetry fullerene configurations (Supplementary Fig. 11), which are adopted from the structures we observed in the MALDI-TOF analysis. Stable structures of different fullerene derivatives are shown in Supplementary Fig. 12a, and among all these cases, $C_{124}$ and $C_{126}$ are found to be the most stable, based on DFT-calculated formation energy differences (Supplementary Fig. 12b). Our MALDI-TOF results reveal that $C_{124}$ and $C_{126}$ have the highest mass fraction in the thermally cycled powder after $C_{60}$, in line with the DFT results. The fraction of $C_{124}$ and $C_{126}$ in the thermally cycled powder, calculated from Supplementary Fig. 8, is estimated to be 5.6% and 2.6%, respectively, while 90% is still $C_{60}$. We note that the same configuration of fullerene at a different m/z value is related to the number of isotopes of $C^{13}$ found in the structure. Also, DFT calculations suggest that increasing the number of $C^{13}$ isotopes leads to an increased formation energy, and therefore to more stable structures (Supplementary Fig. 12c).

We further elucidated the origin of the DOS changes detected via UPS measurements by modeling the perovskite/$C_{60}$ (representative of fresh powder) and perovskite/$C_{124}$ (representative of thermally cycled powder) interfaces. For the perovskite/$C_{60}$ interface, we observed the formation of charge traps near the perovskite band–edge which is attributed to the overlapping of quantum-mechanical wave functions of the perovskite and $C_{60}$, as reported in our earlier work[34]. At that time, we overcame this issue by displacing the fullerene contact with ultrathin (<2 nm) $MgF_2$ interlayers. However, this strategy did not help to reduce the performance losses for fullerene layers deposited via our repeated deposition cycles (Supplementary Fig. 13). This is because interfacing perovskite with $C_{124}$ results in deep trap levels within the band gap as shown in Fig. 3d, which is different than $C_{60}$. By performing the same simulation for different fullerenes ($C_{118}$, $C_{120}$, $C_{122}$, $C_{126}$, and $C_{128}$), we found a similar behavior as $C_{124}$ (Supplementary Fig. 14a). The observed charge traps might be related to formation of multiple bonds between C-I and C-Pb (Supplementary Fig. 14b), in the absence of continuous contact displacer layers (e.g., $CaF_2$, $MgF_2$). Also, in the case of the perovskite/$C_{60}$ model, the delocalization of electrons over the entire $C_{60}$ molecule implies effective vertical electron transport from the perovskite layer to the electrode via $C_{60}$. On the other

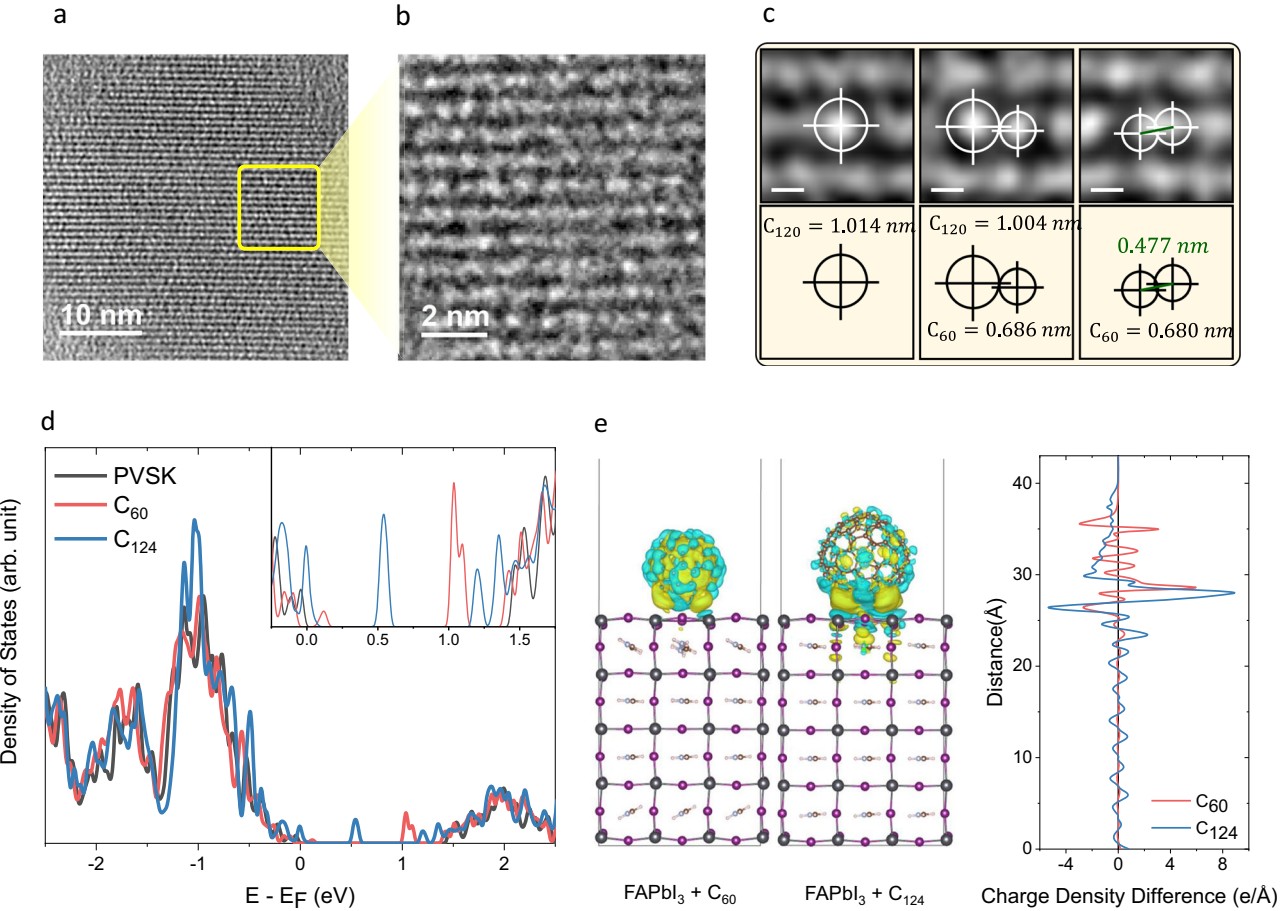

**Fig. 3 | Structural and optoelectronic analysis of as-received $C_{60}$ powders and their films. a** HRTEM image of the highly molecularly crystalline region of thermally cycled $C_{60}$ with low and **b** high magnification. **c** Zoom-in filtered HRTEM images showing possible fullerene derivatives. **d** DOS of pristine perovskite, perovskite/$C_{60}$, and perovskite/$C_{124}$ structures. **e** Charge density difference (CDD) distributions of perovskite/fullerene slabs. The charge delocalization is homogeneously distributed along the $C_{60}$ molecule while the charges in $C_{124}$ are distributed locally just above the surface, which can hinder the charge transfer along the z-axis.

hand, the charges are localized near the perovskite/$C_{124}$ interface, which can hinder the charge transfer across the perovskite/$C_{124}$ interface as shown in Fig. 3e. Therefore, in the case of thermally cycled $C_{60}$, contact displacement does not mitigate performance losses since the $C_{60}$ itself exhibits a defective character.

To understand the possibility of sample storage before the cyclic processing causing performance losses, we conducted an experiment where we compared the performance of half-devices stored in an $N_2$ glove box and vacuum environment (for a duration equivalent to 8 deposition cycles) with that of a fresh half-device that was not stored in the glove box. The completed devices maintained their $V_{oc}$ and $FF$ values without any significant change before and after storage, as illustrated in Supplementary Fig. 15a and b. Consequently, we can conclude that the observed variations are attributed to changes in the fullerene itself.

We further investigated the influence of the $C_{60}$ evaporation heating and cooling cycles on the device performances by performing a single, long deposition cycle, to mimic possible industrial processing. During this process, the substrate shutter was kept closed for a time equivalent to 8 depositions. Note that the number of deposition cycles might be even higher in actual industry processes as additional concerns there will be effective material utilization and throughput. After this period, the shutter was opened for a given time to achieve an identical thickness as in the previous cell fabrication processes. In our study, we observed that the PV parameters of both the long and single

deposition batches closely resembled those of the initial short process cycle (Supplementary Fig. 16a). These findings were verified using two separate batches, which yielded consistent results for both the fresh deposition and the long thermal cycle of $C_{60}$. Interestingly, the rising crucible temperature during the extended cycle (Supplementary Fig. 16b) suggests that temperature increase is not the cause of the performance losses. Instead, we hypothesize that coalescence occurring in or on the fullerene powders during source cooling in half-device exchanges might be responsible for these outcomes.

### Repeated deposition cycles with sublimed $C_{60}$

In search of a solution to overcome $C_{60}$ coalescence, we further hypothesized that impurities in the as-received powders might be responsible for the dimer formation[38]. To verify this, we performed high-performance liquid chromatography (HPLC) analysis and we found that the as-received $C_{60}$ has a purity of ~99.75-99.8% which can contain impurities or by-products of $C_{60}$ oxide and $C_{70}$, in addition to the possibility of the presence of $C_{60}$-dimer as shown in Supplementary Fig. 17. We further purified the as-received $C_{60}$ powders via vacuum thermal gradient sublimation (see Methods), which resulted in enhancing the purity of $C_{60}$ to 99.95% as verified by HPLC analysis in Supplementary Fig. 18 (henceforth referred to as "sublimed"). After purification, we performed repeated deposition cycles similar to the as-received series of experiments. Interestingly, the color of thermally cycled sublimed $C_{60}$ remained visually black even after 8 deposition

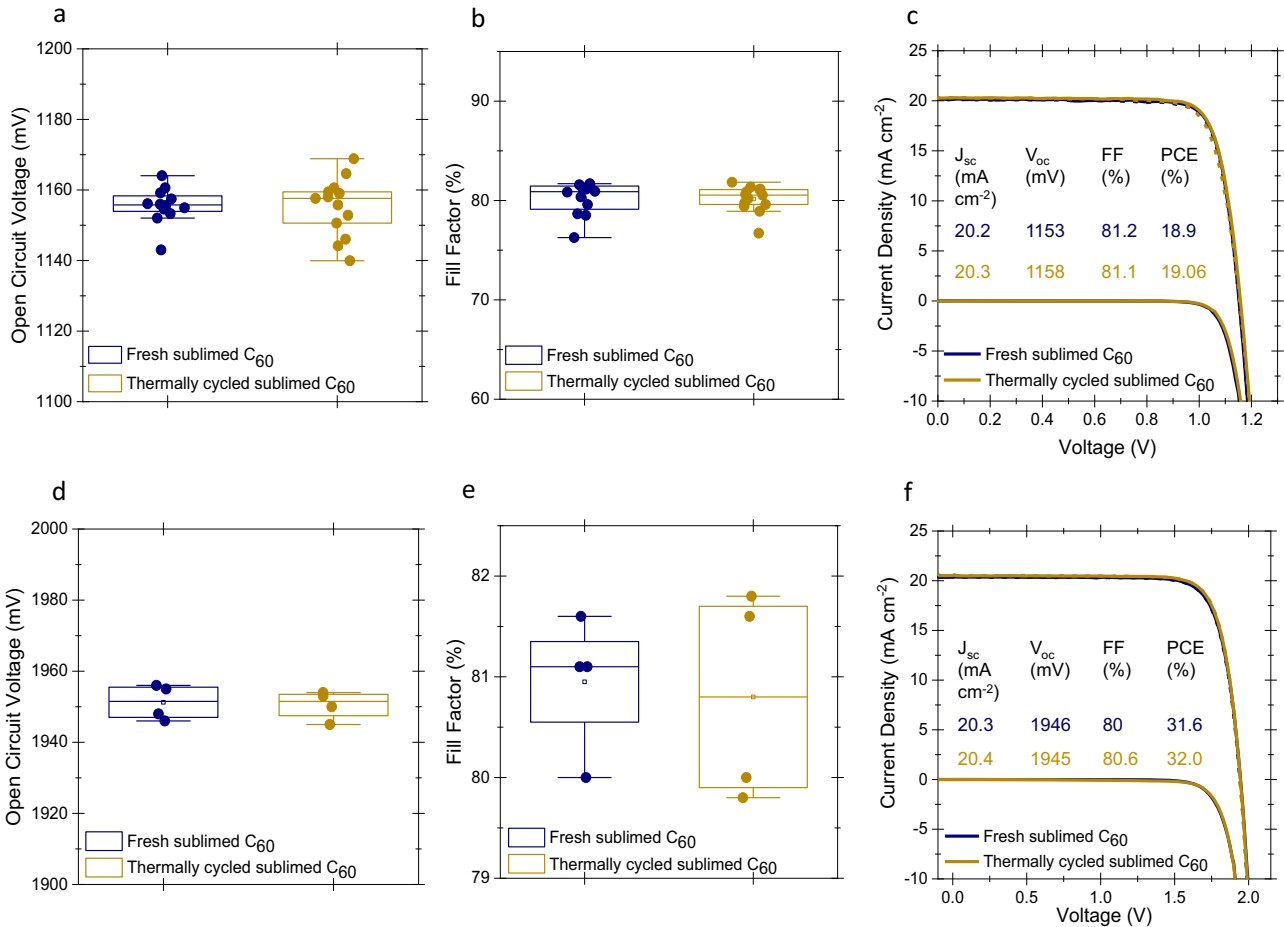

**Fig. 4 | Photovoltaic performances with contacts using sublimed C$_{60}$.** **a** Statistical distribution of $V_{oc}$ and **b** $FF$ of single-junction solar cells, together with **c** representative $J$–$V$ curves. **d** Statistical distribution of $V_{oc}$ and **e** $FF$ of perovskite/silicon tandem solar cells, together with **f** representative $J$-$V$ curves.

cycles as shown in Supplementary Fig. 19a. Furthermore, in contrast to the as-received material, the $T_c$ profile remained unchanged when the number of deposition cycles increased. Also, the maximum $T_c$ remained lower than that of the as-received C$_{60}$ (see temperature profile in Supplementary Fig. 19b), which was confirmed by thermo-gravimetric analysis (TGA) under nitrogen (Supplementary Fig. 20).

We studied the sublimed C$_{60}$ layers with the same set of characterization techniques to observe the influence on the film quality. QFLS analysis of half-device stacks (NiO$_x$/MeO-2PACz/perovskite/sublimed C$_{60}$) still exhibited losses after 8 deposition cycles. However, compared with that of as-received C$_{60}$ as shown in Supplementary Fig. 21a, the loss after each cycle is only ~2 meV. To understand the possible occurrence of defect states in the perovskite band gap after repeated deposition cycles of sublimed C$_{60}$, half stacks of quartz/perovskite/sublimed C$_{60}$ were characterized by PDS. Here, the measured PDS results were almost identical for fresh and thermally cycled sublimed C$_{60}$, as shown in Supplementary Fig. 21b, c. Also, WFs and HOMOs of thermally cycled sublimed C$_{60}$ remained almost unchanged (Supplementary Fig. 21d, e). Importantly, we detected from XPS analysis the same carbon content but an absence of oxygen in the fresh and thermally cycled sublimed C$_{60}$ samples (Supplementary Fig. 22). MALDI-TOF analysis of fresh and thermally cycled sublimed C$_{60}$ powders exhibited a very low oxygen percentage, which is around 0.6% as shown in Supplementary Fig. 23a, by showing peaks at 1416, 1417, and 1418 m/z of C$_{118}$. The PL analysis of powders also shows formation of a shoulder at 835 nm, which is less pronounced with sublimed C$_{60}$ (Supplementary Fig. 23b). This suggests the presence of fewer impurities. This peak is associated with phonon replicas in literature[42].

Our characterization set revealed that further purification of as-received C$_{60}$ through a sublimation process is a promising solution to retain the optoelectronic properties of C$_{60}$ after repeated deposition cycles. So, we studied sublimed C$_{60}$ on solar cells with the identical stacks given in Fig. 1a. By analyzing the $J$-$V$ curves of the fabricated devices, we found the average of $V_{oc}$ and $FF$ of fabricated devices leveled off as shown in Supplementary Fig. 24. Even fully cooling down this powder to room temperature and using it again did not affect the average device performance (Fig. 4a, b). The corresponding $J$-$V$ curves are shown in Fig. 4c. Also, the performances of the perovskite/silicon tandem solar cells were unaffected by multiple thermal cycles if sublimed C$_{60}$ was used (Fig. 4d, e). Corresponding $J$-$V$ curves and EQE graphs of tandem devices are shown in Fig. 4f and Supplementary Fig. 25, respectively. One of our unencapsulated perovskite/silicon tandem devices using sublimed C$_{60}$ showed a certified PCE of 30.90% by Fraunhofer ISE CalLab as shown in Supplementary Fig. 26. We note that using sublimed C$_{60}$ provides a quite marginal $V_{oc}$ enhancement (<10 mV), however, it gives reproducible results in repeated processes, which will be critical to enable the industrial adoption of this technology.

## Discussion

The objective of this work was to investigate the feasibility of industrial processing of C$_{60}$ contact layers in perovskite-based solar cells through repeated thermal evaporation depositions. To accomplish this, we employed an evaporation system with a rotating substrate and automated load lock for half-device exchange, without changing the source powders until the end of the batch. After multiple repeated

processes, we assessed film quality and solar cell performances and found that if 99.75% pure as-received $C_{60}$ powders (as-received) are used, the powders should be renewed for each deposition cycle (in case heating and cooling are involved) to avoid a decrease in device performance.

However, in high throughput lines, this demands frequent source replenishment to maintain device performance and may lead to sub-optimal materials utilization and added costs. We identified this as a potential problem for industrial processing of perovskite-based solar cells, whether in their single-junction or multijunction (e.g., perovskite/silicon, perovskite/perovskite, or three-junction) implementations. The decrease in device performance is thought to be caused by the coalescence of $C_{60}$ to larger fullerene derivatives, initiated by the presence of oxygen. Notably, no decrease in device performance was observed when performing a single, long deposition cycle equivalent to the duration of 8 deposition cycles involving heating and cooling.

Our work demonstrates that using 99.95% pure $C_{60}$ powders, purified via vacuum thermal gradient sublimation, can address this issue and enables continuous processing of $C_{60}$ contact layers without systematic performance losses. We hypothesized that some oxygen was already enclosed in the crystal structure of 99.75% pure as-received $C_{60}$. Formation of nearly perfect face-centered-cubic (fcc) crystals in the sublimation process not only eliminates existing oxygen content from voids in the regular crystal lattice but also eliminates defects to prevent subsequent oxygen uptake except for surface adsorption which is easily reversible during degassing prior to the start of the vapor deposition process. Another possibility, beyond sublimation, is the development of processes to produce oxygen-free, and/or oxygen uptake-resistant $C_{60}$ without the need for sublimation.

Towards the industrialization of $C_{60}$ contacts, weak mechanical adhesion[43], instability under field operation[44], and non-ideal contact behavior have been reported previously[34]. Solving these pressing issues is urgent, as commercial PV modules with $C_{60}$ contacts are already in sight. In addition to the potential of using sublimed $C_{60}$, this work provides valuable guidance for the preparation of fullerenes that are more suitable, particularly oxygen-free, for commercial perovskite-based solar cell processing with high yield.

## Methods

### Vacuum thermal gradient sublimation
Commercially purchased $C_{60}$ (Nano-C) was sublimed via a three-zone tube furnace (Lindberg Blue M, Model number STF55346COMC-1), with quartz collection tubes in it. Prior to use, all quartz pieces were sequentially cleaned by acetone, IPA, and piranha solution (3:1 sulfuric acid and hydrogen peroxide), followed by a quartz bake-out process at 500 °C in the furnace. According to the materials' vapor pressure, the temperatures of the three zones were set to 570 °C, 395 °C, and 340 °C, respectively, to sublime the $C_{60}$ powders and create the proper temperature gradient for pure $C_{60}$ crystals and impurities to be separated. During the purification, the tube environment was kept at 10 Torr by a vacuum pump (Oerlikon Leybold vacuum 501591A1000 Turbolab 80). A constant nitrogen gas flow (20 SCCM flow rate, controlled by Mass Flow Controller SFC5400 LA6UAV) was applied to assist material transport along the tube. The final yield was in the range of 70–80%.

### Single-junction solar cell fabrication
17 nm of $NiO_x$ was sputtered on patterned ITO (Xin Yan Technologies, 15 ohm/sq) at room temperature using a NiO target (Plasmaterials, 99.9%) in argon atmosphere. To passivate defects on the surface of $NiO_x$, potassium chloride, KCl (3.5 mg/mL in water) was spin coated on the $NiO_x$ at 4000 rpm and then annealed at 120 °C for 10 min. Then 1 mg/ml of MeO-2PACz (99 %, TCI) was spin-coated on $NiO_x$ at 5000 rpm followed by annealing at 100 °C for 10 min. MeO-2PACz was dynamic washed by DMF to obtain a monolayer and annealed at

100 °C. $NiO_x$/MeO-2PACz is considered as an optimal HTL for hybrid-processed perovskite layer. The inorganic scaffold of 240 nm $PbI_2$-CsBr was evaporated on MeO-2PACz/ $NiO_x$ with rate of 1 and 0.1 A/s for $PbI_2$ and CsBr, respectively. The final perovskite layer was obtained after conversion of the inorganic scaffold by dynamic dripping 0.65 M ethanol solution of FAI/FABr/MACl with a molar ratio of 6.2/2.07/1.0, respectively at 4000 rpm for 40 s. The crystallization of the perovskite layer was obtained after annealing at 150 °C for 30 min in ambient air. The half devices with the stack of ITO/$NiO_x$/MeO-2PACz/perovskite were split into 8 batches. 25 nm of $C_{60}$ was evaporated on each batch from the same $C_{60}$ without changing or adding additional material. Before the deposition, 310 mg of $C_{60}$ powder was contained in the crucible. Between the processes, the material in the crucible did not cool down below 240 °C. The devices are finalized with 6 nm BCP and 120 nm Ag contact again with thermal evaporation techniques.

### Perovskite/silicon tandem solar cells fabrication
c-Si bottom cells were prepared as in our previous work[34]. The bottom cells were subjected to UV-Ozone treatment for 10 min. After that, 2PACz was applied via 1 mg/mL solution in ethanol by spin-coating on ITO-coated substrates at 5000 rpm for 30 s, followed by drying at 100 °C for 10 min. We utilized 2PACz as a hole extraction layer in solution-processed perovskite layer following our tandem solar cell baseline procedure[5]. 1.7 M $Cs_{0.05}FA_{0.8}MA_{0.15}Pb(I_{0.755}Br_{0.255})_3$ perovskite precursor solution was prepared by dissolving a mixture of FAI, MABr, CsI, $PbI_2$, and $PbBr_2$ in a solvent mixture of DMF and DMSO with a volume ratio of 4:1, and spin-coated at two steps (2000 rpm for 45 s and 7000 rpm for 10 s) chlorobenzene of 200 µL was dropped in the center of the substrates 12 s before the end of the spin-coating process. The substrates were annealed on a hotplate at 100 °C for 15 min. After perovskite deposition, -1 nm metal fluorides ($MgF_2$), and 15 nm $C_{60}$ were subsequently deposited by thermal evaporation. 10 nm $SnO_2$ was then deposited by atomic layer deposition (ALD) using a Picosun system at 100 °C. Tetrakis(dimethylamino)tin(IV), TDMASn precursor source was at 80 °C and $H_2O$ source was at 25 °C. The pulse and purge time for TDMASn was 1.6 and 5.0 s with a 90 sccm carrier gas of nitrogen, for $H_2O$ is 1.0 and 5.0 s with 90 sccm $N_2$, 85 cycles were used. 40 nm IZO was sputtered from a 3-inch IZO ceramic target on top of the $SnO_2$ through a shadow mask. Ag finger with a thickness of 500 nm was thermally evaporated using a shadow mask. Finally, 100 nm $MgF_2$ was thermally evaporated as an anti-reflection layer. The evaporation rate and thickness of each experiment were monitored by quartz crystal microbalance sensors.

### Solar cell characterizations
Current density-voltage ($J$–$V$) curves were measured by Keithley 2400 source meter under calibrated AM1.5 illumination using a solar simulator (Abet Technologies Sun 3000 Solar Simulator) inside a nitrogen-purged glovebox. The range of the forward scan rate and reverse scan rate is −0.1 V to 1.2 V and 1.2 V to −0.1 V, respectively for single junction cell with active area 0.1 $cm^2$. For tandem cell the range of the forward scan rate and reverse scan rate was −0.1 V to 2.0 V and 2.0 V to −0.1 V, respectively (area of in house measurement is 1.04 $cm^2$). Both scans were carried out with scan step 50 mV/s. For the tandem solar cells, Wavelabs Sinus 220 LED-based solar simulator was utilized.

### Mobility measurements via field effect transistor
Glass substrate with pre-patterned source and drain Au electrodes with a channel length of 40 µm and channel width of 1000 µm were used. $C_{60}$ layers were thermally evaporated with a thermal evaporator through a shadow mask. Cytop dielectric was spin-coated at 2000 rpm for 60 s on top and annealed at 60 °C for 90 min. The aluminum top metal gate was evaporated using a shadow mask to complete the device architecture. The measurements were performed via Agilent B2912A source/measure unit (SMU) in a glove box.

**Transmission electron microscopy**

TEM sample was prepared by dispersing thermally cycled $C_{60}$ in chlorobenzene and drop casting on a holey carbon support film TEM grid. TEM was performed with Cs spherical aberration image corrected Thermofisher Titan 60-300 Cubed TEM, TEM microscope operated at low acceleration voltage (80 kV). Gatan Digital Micrograph was used to process the data.

**Hyperspectral PL Imaging and QFLS.** To mimic *p-i-n* device structure, perovskite films were deposited on HTL/ITO films using the same method as for device fabrication. Then, fresh and thermally cycled $C_{60}$ films were evaporated on top of the perovskite films, respectively. Absolute PL of encapsulated half-finished devices were collected using a hyperspectral imaging system coupled to a microscope with 2 nm spectral resolution (Photon etc. IMA) and stacks were excited with a 405 nm laser at around 1 sun illumination[4]. The collected data were analyzed by home-built MatLab code using modified Würfel's generalized Plank law to acquire QFLS, $\Delta\mu$[45,46].

**XPS and UPS.** All $C_{60}$ films were deposited on ITO substrates. The electron spectroscopy measurements of ultraviolet photoelectron spectroscopy (UPS) and X-ray photoelectron spectroscopy (XPS) were carried out with a UHV ScientaOmicron system at 5E-10 mbar. UPS measurements were performed on the samples using a vacuum UV source (focus) operating at 200 mA and 2E-2 mbar He pressure to increase the fraction of the He(II) line (40.8 eV). The kinetic energy range scanned was increased to 30–50 eV and the spectra were adjusted accordingly. The photoelectrons were collected at an angle of 80° between the sample and analyzer, with a normal electron takeoff angle. The constant analyzer pass energy (CAE) was 5 eV for the valence band region and for the secondary electron cutoff SECO. Spectra were adjusted by a scaling factor to provide a suitable comparison. XPS was carried out in the same spectrometer, equipped with a monochromatic Al Ka X-ray Omicron XM1000 X-ray source (hν = 1486.6 eV) operating at a power of 390 W. The survey and high-resolution spectra were collected at a CAE of 50 and 15 eV, respectively. The spectra were analyzed with Casa XPS software. The individual peak envelopes were fitted by a Gaussian–Lorentzian (GL30) using a Tougaard-based background function.

**DFT.** DFT is used to carry out quantum mechanical calculations in the gas phase, as implemented in Gaussian 09[47]. wB97-XD/6-31 G(d) functional and basis set is used to optimize all gas phase geometries. Possible high symmetry fullerene configurations within $n = 118$–$128$ were initially generated using the CaGe code[48] and each structure if DFT optimized. Final electronic energies are used to collect the most stable geometry for each fullerene type. The formation energy of each fullerene (relative to that of $C_{60}$) is then predicted from

$$\Delta E = E_n/n - E_{C_{60}}/60 \qquad (1)$$

where $E_{C_{60}}$ and $E_n$ is the gas phase energy of $C_{60}$ and $C_n$. For the bulk phases, we performed first-principles calculations based on DFT by a plane-wave basis set and the projected augmented wave method, as implemented in the VASP package[49,50]. Perdew−Burke−Ernzerhof (PBE) functional with a generalized-gradient approximation is employed for the exchange-correlation functional in all geometry optimization and self-consistent field calculations. Fresh[51] A $4 \times 4 \times 1$ gamma-centered k-mesh and a plane wave basis with a 400 eV cutoff energy is used for the geometry optimizations. We first allowed relaxing the atomic positions and cell volumes using a conjugate gradient algorithm, until all residual forces are <0.02 eV/Å. Continuation single-point energy calculations on the optimized geometries were performed to create the charge density difference graphs. We insert a vacuum slab with thickness of $10 - 15$ Å between the periodic slab-molecule structure along z direction.

**PDS.** The photothermal deflection spectroscopy (PDS) measurements were performed using a home-built PDS setup. Samples were immersed in a chemically inert liquid perfluorohexane $C_6F_{14}$ (FC72). The light from the 150 W Xe lamp was sent through a monochromator equipped with gratings blazed at 300, 750, and 1250 nm, and modulated by a mechanical chopper operating at a frequency of 5 Hz. A probe beam passing parallel to the sample surface is deviated by a change of the refraction index of the FC72 and detected by a four-quadrant detector.

**XRD.** XRD data was collected by using Bruker D2 Phaser diffractometer (Cu-$K_{\alpha1}$ radiation, $\lambda = 1.5406$ Å) at room temperature from 5° to 50° (2θ) with a scan speed of 3°/min.

**HPLC.** High-pressure liquid chromatography (HPLC) was carried out with an Agilent 1100 instrument equipped with a variable wavelength detector (VWD). An analytical Cosmosil Buckprep column (Nacalai, 250 mm × 4.6 mm I.D.) was used with 1 mL/min of toluene for elution and detection at 330 nm. $C_{60}$ samples were dissolved in toluene at a concentration of 0.25 mg/mL and 20 μm are injected into the instrument.

**TGA.** Thermogravimetric analysis (TGA) was done in a nitrogen atmosphere using a TA Discovery TGA 5500 instrument after careful purging. The heat rate was 10 K/min from room temperature to 1000 °C.

**Reporting summary**

Further information on research design is available in the Nature Portfolio Reporting Summary linked to this article.

## Data availability

All generated data in this study is available in Supplementary Information.

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

## Acknowledgements

This work was supported by the King Abdullah University of Science and Technology (KAUST) Office of Sponsored Research (OSR) under Award No: OSR-CARF URF/1/3079-33-01, KAUST OSR-CRG 400 RF/1/3383, KAUST OSR-CRG2018-3737. M.L. and A.V. acknowledge Czech Ministry of Education, Youth and Sports grant no. LUASK 22202 and the use of the CzechNanoLab research infrastructure (LM2018110). T.D.A. acknowledges the baseline funds provided by the King Abdullah University of

Science and Technology. Computing resources used in this work were provided by the National Center for High-Performance Computing of Turkey (UHeM) under the grant number of 1015902023. The authors thank Shruti Sarwade for her contribution to the c-Si bottom cell preparation and Dr. Safiye Sag Erdem for fruitful discussions on DFT analysis.

## Author contributions

A.A.S. and E.A. conceived the idea and wrote the manuscript. A.A.S., B.K.Y., R.A., and J.L. fabricated and characterized the single-junction perovskite solar cells. E.U. and E.A. fabricated and characterized the perovskite/silicon tandem solar cells. B.V. performed TEM measurements and related data analysis. M.B., T.G.A., and A.R. fabricated silicon bottom cells. C.D. and I.Y. performed the DFT calculations and related data analysis. A.A.S. and H.B. contributed to the MALDI-TOF data analysis. A.V. and M.L. performed the PDS measurements and related data analysis. S.M. and T.D.A. fabricated the OFET devices and extracted the data. A.P. explained the OFET and mobility measurements. P.D. performed the XPS and UPS measurements and related data analysis. Z.X., H.M.J., M.G. and B.P.R. performed the sublimation process of as-received $C_{60}$. H.R. and E.J. performed TGA and HPLC analyses. All authors contributed to the writing. E.A. and S.D.W. supervised the project and S.D.W. secured the funding.

## Competing interests

The authors declare no competing interests.
