## [Peer Review File · Nature Communications]

Sublimed C60 for efficient and repeatable perovskite-based solar cellsREVIEWER COMMENTS

Reviewer #1 (Remarks to the Author):

This manuscript reports the influence of C60 aggregation on perovskite solar cells' performance and the reproducibility of p-i-n device configuration useful for silicon/perovskite tandem solar cells. The findings highlight the evolution of C60 thin films' electronic quality through repeated evaporation processes and its impact on device performance. By using purified C60, it's possible to mitigate the negative effects and maintain device performance for perovskite-based photovoltaic technologies. The study contributes to a better understanding of the role of C60 thin films in electronic devices and offers practical insights for optimizing their performance. The findings are in the interest of Nature Communications' readership, and we recommend publication after addressing the following queries.

- (1) The authors report purified (sublimed) C60 avoids aggregation issues, and device performance remains unaffected even after repeated deposition cycles. What are the impurities that are removed by sublimation?
- (2) The electron mobilities (μ_e) of the 1st deposition (0.083 cm²/V s) are higher than the 8th deposition (0.034 cm²/V s), implying the structural changes of the C60 layer, which is important to characterize by spectroscopic techniques.
- (3) The presence of an emission shoulder in the uncontaminated sample, particularly around 800 nm, becomes more pronounced in the emission peak of thermally cycled powders at 835 nm. This is in addition to the primary peak for pure C60 situated at 742 nm, underscoring that even the initial sample exhibits certain issues.

Reviewer #2 (Remarks to the Author):

This manuscript mainly reports that low-purity C60 will affect batch repeatability of device performance due to oxygen-induced coalescence during repeated thermal evaporation processes. The coalescence of C60 leads to the formation of deep states, increasing additional nonradiative recombination, thus resulting in a decrease in device performance. After sublimation and purification of C60, the purity has been improved to 99.95%, effectively solving the problem of repeatability, and achieving a certified power conversion efficiency of 30.9% for perovskite-silicon tandem solar cells. This work provides unique insights into the fabrication of C60 layers in perovskite-based devices, so I recommend publishing it in Nature Communication. The following questions are suggested for improving the manuscript.

1. The available evidence fully demonstrates that the C60 powder collected from the crucible after the 8th cycle produces a large change, but there is no direct evidence for the difference of C60 evaporated onto the perovskite from different batches. Direct analysis of the changes in the evaporated C60 rather than the changes in the composition of the residual C60 can better help us understand this phenomenon.
2. What is the relationship between material purity and the electronic defects? What kinds of impurities will cause deep level electronic trap states and how these coalesced C60 produce these trap states? Does this caused by energy levels or other factors?
3. On the other hand, for as-received C60, is there any difference in the morphology and thickness of C60 layers in different batches?
4. If the evaporation equipment is integrated in the glove box and there is no oxygen introduction between different batches, is there a problem with the repeatability of the device? Similarly, what would happen if each batch was cooled to room temperature before oxygen was introduced. Can this solve the problem of oxygen-induced coalescence?
5. Although the efficiency of tandem solar cells is very high, the efficiency of single-junction perovskite solar cells is relatively low, especially the open circuit voltage, even after introduce CaF₂ as interlayer in figure s11. How can such inefficient single-junction devices achieve efficient tandem devices?
6. In figure 1d, the HTL is NiO/MeO-2PACz. However, in Perovskite/silicon tandem solar cells fabrication, the HTL is 2PACz. Is there any difference for these two hole-selective materials?

Reviewer #3 (Remarks to the Author):

This paper aims to elucidate the effects of repeated C60 evaporation on single-junction and silicon/perovskite tandem solar cell devices, with a particular focus on the high efficiency (31%) achieved in the silicon/perovskite tandem configuration. The study appears to have meticulously conducted film analyses and calculations concerning the reuse of C60, as depicted in Fig. 2 and Fig. 3. If the conclusions of this study prove valid, they could offer invaluable insights to numerous research groups employing C60 and industrial institutions endeavoring to commercialize perovskite single-junction or tandem devices with a p-i-n configuration. However, a substantial set of questions and concerns arise concerning the primary aspect of this paper - the device data intended to substantiate the main claims of the authors. Addressing these concerns is imperative for mandatory revision, as outlined below, before I could recommend the publication of this manuscript in Nature Communications.

Firstly, the pivotal data that supports the author's conclusions, as presented in Fig. 1, appears insufficient to robustly validate their claims. If I understand correctly, for single-junction data, the authors fabricated the half-device (ITO/HTL/Perovskite) within the same batch. Subsequently, they evaporated C60 for the first set of substrates, stored them in a load lock (under low-vacuum conditions), evaporated C60 for the

second set of substrates from the same batch as the first C60 devices set, similarly stored them in a load lock, and repeated this process up to the eighth C60 set before finally introducing BCP/Ag and conducting the experiment. This approach seems appropriate for assessing the influence of the number of C60 depositions. However, at this juncture, I'm curious that how many times the authors repeated this experiment, and the specifics of the experimental design. To ensure that the repeated use of C60 genuinely affects device performance, it is crucial for the authors to replicate this process in separate experiments at least 3-4 times (not for 3-4 substrates at one time) using the same batches of NiOx/SAM/Perovskite layers. Explicitly detailing the steps taken to ensure controlled conditions that exclusively account for the impact of C60 reuse, while excluding other potential performance variations, is essential to convince the readers.

Additionally, measures to mitigate variables stemming from the time spent in the load lock need to be established (e.g., comparing substrates stored without using the load lock and substrates stored in the load lock for the same duration as the deposition times for the 1st to 8th C60 experiments, followed by experiments using fresh C60 to verify consistency).

Furthermore, a major factor contributing to the lack of persuasiveness in the device data is the behavior observed in the Fig. 1 graph. A drop is evident between the first and second data points, followed by almost consistent values between the 2nd and 8th data points, with a subtle decrease at the end. This pattern does not match well with the QFLS data in Figure 2. Efficiency remains nearly constant between the 2nd and 7th data points (excluding the 1st and 8th). It is imperative to elucidate whether the divergent efficiencies at the 1st and 8th data points are coincidental or genuinely attributed to the reuse of C60. The tandem data in Fig. 1 does not exhibit this discrepancy as prominently. The J-V curves in the tandem data are almost superimposed, making it challenging for readers to accept the efficiency drop posited by the authors. While a numerical efficiency drop of 0.6% presented, it's a small value; it might be more compelling to represent this reduction through the distribution illustrated in the supplementary materials.

I acknowledge that repeating the entire experiment for all conditions is a considerable undertaking. Therefore, I recommend focusing on replicating 2-3 types of devices: 1) using fresh C60, and 2) applying 8 cycles of heating and cooling for C60(w/o fabricating device), followed by fabricating the device repeatedly using the same batch of perovskite layers, while verifying the absence of any load lock storage influence. Additionally, it could be beneficial to introduce one more condition in the middle, involving 4 cycles of heating and cooling, if feasible for the authors.

This type of data could be statistically integrated into Figure 1. If the authors' claims hold true and similar results are achieved, the disparities could be more explicitly displayed on the Fig. 1 graph, featuring an ample number of data points. Moreover, with the careful specification of the controlled conditions mentioned earlier, readers are likely to find greater resonance with the authors' assertions alongside the improved trends demonstrated by the enhanced Fig. 1 graph.

REVIEWER COMMENTS

Reviewer #1

This manuscript reports the influence of C₆₀ aggregation on perovskite solar cells' performance and the reproducibility of p-i-n device configuration useful for silicon/perovskite tandem solar cells. The findings highlight the evolution of C₆₀ thin films' electronic quality through repeated evaporation processes and its impact on device performance. By using purified C₆₀, it's possible to mitigate the negative effects and maintain device performance for perovskite-based photovoltaic technologies. The study contributes to a better understanding of the role of C₆₀ thin films in electronic devices and offers practical insights for optimizing their performance. The findings are in the interest of Nature Communications' readership, and we recommend publication after addressing the following queries.

We thank the reviewer for the positive feedback and recommendation for publication. We appreciate the reviewer's assessment of the paper's significance and the clarity of the presented mechanism. In our response letter, we carefully considered the reviewer's suggested changes.

(1) The authors report purified (sublimed) C₆₀ avoids aggregation issues, and device performance remains unaffected even after repeated deposition cycles. What are the impurities that are removed by sublimation?

In our initial submission, we provided in Figure S14 (has been updated in the Supplementary Information as Figure S17) the high-performance liquid chromatography (HPLC) chart which confirms that as-received C₆₀ has a purity of 99.75-99.8 % and includes C₆₀ oxide, C₇₀, and C₆₀ dimers as well as *unidentified* impurities eluting before C₆₀. While not definitively confirmed, such relatively early elution while absorbing at 330 nm could point to unidentified impurities being fullerene fragments or fullerene adducts. After sublimation, the purity of sublimed C₆₀ increased to 99.95% (as shown in Figure S18) mainly by removing the unidentified impurities from the powder. We updated Figure S17 and S18 with better integrations.

(2) The electron mobilities (μ_e) of the 1st deposition (0.083 cm²/V s) are higher than the 8th deposition (0.034 cm²/V s), implying the structural changes of the C₆₀ layer, which is important to characterize by spectroscopic techniques.

Thanks for raising this point to support our manuscript. This question motivated us to do more analysis on thin films, other than powders. To respond to the reviewer's question, we carried out a new MALDI-TOF mass spectroscopy analysis for newly prepared fresh (1st cycle) and thermally cycled (>8 cycles) as-received (non-sublimed C₆₀) C₆₀ thin films. For this, we deposited layers on large clean glass substrates and scratched them with clean blades, and we performed the measurements on collected powders.

As shown in Figure R1, the MALDI-TOF analysis indicates the presence of the peaks at 1322, 1345, 1369, 1392, and 1417 m/z in both C₆₀ thin films with different peak intensities, which are assigned to C₁₁₀, C₁₁₂, C₁₁₄, C₁₁₆, and C₁₁₈, respectively. Importantly, the peaks of thermally cycled thin film are stronger than that of fresh thin film, which confirms the concentration of the higher molecular weight fullerenes in thermally cycled thin film is higher than that of fresh thin film.

Interestingly, the fresh C_{60} thin film lacked the peak at 1441 m/z for C_{120} , which appeared in the thermally cycled thin film. Figure R1 has been added to the main text as Figure 2f.

Figure R1: MALDI-TOF analysis of fresh and thermally cycled as-received C_{60} thin films.

(3) The presence of an emission shoulder in the uncontaminated sample, particularly around 800 nm, becomes more pronounced in the emission peak of thermally cycled powders at 835 nm. This is in addition to the primary peak for pure C_{60} situated at 742 nm, underscoring that even the initial sample exhibits certain issues.

We understand that the reviewer refers to the PL spectra acquired from powders shown in Figure S6b. Indeed, we have a shoulder even for the as-received C_{60} powders (before repeated deposition), and obviously, we have some other impurities in these samples as reported in the HPLC analysis in Figure S17.

To understand the relation of impurities on the peak located around 835 nm, we now analyzed the *sublimed* C_{60} powder via PL, in addition to our previous analysis on as-received C_{60} powder. As shown in Figure R2, the sublimed C_{60} showed a weaker peak at 835 nm, which implies even after purification still the peak exists and that the peak at 835 nm becomes stronger with an increase in the impurity content of C_{60} . Figure R2 has been added to the Supplementary Information as Figure S23b.

In literature, the PL peak at 742 nm is assigned to radiative recombination, while the peak at 835 nm is attributed to phonon replicas, which is an inherent feature of C_{60} (Solid State Communications, Vol. 98, No. 9, 853-858, 1996).

Figure R2: PL of the fresh and thermally cycled as-received and sublimed C_{60} .

Reviewer #2

This manuscript mainly reports that low-purity C_{60} will affect batch repeatability of device performance due to oxygen-induced coalescence during repeated thermal evaporation processes. The coalescence of C_{60} leads to the formation of deep states, increasing additional nonradiative recombination, thus resulting in a decrease in device performance. After sublimation and purification of C_{60} , the purity has been improved to 99.95%, effectively solving the problem of repeatability, and achieving a certified power conversion efficiency of 30.9% for perovskite-silicon tandem solar cells. This work provides unique insights into the fabrication of C_{60} layers in perovskite-based devices, so I recommend publishing it in Nature Communication. The following questions are suggested for improving the manuscript.

We extend our gratitude to the reviewer for recognizing the distinctiveness and depth of our contribution to the perovskite-based solar cells field. We appreciate the recommendation for the publication of our work in Nature Communications. With this opportunity, we improved the quality of our article further by addressing the queries of the reviewer.

1. The available evidence fully demonstrates that the C_{60} powder collected from the crucible after the 8th cycle produces a large change, but there is no direct evidence for the difference of C_{60} evaporated onto the perovskite from different batches. Direct analysis of the changes in the evaporated C_{60} rather than the changes in the composition of the residual C_{60} can better help us understand this phenomenon.

We thank the reviewer for raising this critical point. This question implies the same manner as R1's 2nd question. We kindly request the reviewer to refer to that section.

2. What is the relationship between material purity and the electronic defects? What kinds of impurities will cause deep level electronic trap states and how these coalesced C_{60} produce these trap states? Does this caused by energy levels or other factors?

Regarding the purity level of the materials; according to the HPLC analysis, we observed C_{60} oxide, C_{70} and C_{60} dimer, and unknown impurities eluting before C_{60} . These unknown impurities, possibly fullerene fragments and/or fullerene adducts, are expected to have electronic properties that are significantly different from those of C_{60} and could therefore negatively impact electron transport.

However, we need to keep in mind the effect of the vapor deposition process of C_{60} . The sublimation temperatures of possibly present fullerene fragments can be expected to be significantly lower than that of C_{60} and therefore such fullerene fragments are unlikely to be deposited on the substrate at the process conditions used here (it might be evaporated during soaking time when the substrate shutter was closed, which means It can present in the films). In that concern, fullerene adducts usually decompose, often reforming C_{60} , under the conditions of vapor deposition (see e.g., Giovane et al., Kinetic Stability of the C_{60} -Cyclopentadiene Diels-Alder Adduct. *J. Phys. Chem.* **1993**, *97*, 8560-8561.). This explains the very similar device performance in the first vapor-deposition cycle for both as-received and sublimed C_{60} powders.

From ultraviolet photoelectron spectroscopy (UPS) analysis, which was already given in Figure 2e (main text) in Figure S6a, we have seen that the states close to the highest occupied molecular orbital (HOMO), or in other words valance band maximum (VBM) show some changes after thermal cycling of the as-received powders but we have not seen any additional states close to the band edge. From PDS analysis, we observe that the C_{60} impurities directly cause nonradiative recombination, which leads to loss in V_{oc} . With our analysis, we demonstrated that the recombination occurs on the perovskite surface. This means that either the defects are created on the PK/ C_{60} interface or the defects are directly in the C_{60} layer. The DFT calculations show deep electronic states at the degraded C_{60} /PK interface, which strongly supports the creation of electronic defects by C_{60} impurities.

3. On the other hand, for as-received C_{60} , is there any difference in the morphology and thickness of C_{60} layers in different batches?

Upon the reviewer's request, we investigated the surface morphology of fresh and thermally cycled as-received C_{60} films was investigated by using atomic force microscopy (AFM). There is no significant change in the morphology of fresh and thermally cycled as-received C_{60} films as shown in Figure R3 a and b, respectively. In addition, the root mean square (RMS) of fresh as-received C_{60} film is 1.2 nm, while the RMS of thermally cycled as-received C_{60} film is 1.47 nm. The thickness of C_{60} film is controlled during the deposition by quartz crystal microbalance (QCM). Therefore, the thickness of fresh and thermally cycled C_{60} films was controlled to reach 20 nm. Figure R3 has been added to the Supplementary Information as Figure S9.

Figure R3: AFM of (a) fresh and (b) thermally cycled as-received C_{60} films.

4. If the evaporation equipment is integrated in the glove box and there is no oxygen introduction between different batches, is there a problem with the repeatability of the device? Similarly, what would happen if each batch was cooled to room temperature before oxygen was introduced. Can this solve the problem of oxygen-induced coalescence?

Our evaporation tool facilitates sample exchange within a vacuum-sealed load lock, as detailed in the Methods section. This load lock is also interconnected with a glove box. During the processing phase, the deposition chamber maintains an ultra-low vacuum level of approximately $1E-7$ Torr, whereas the load lock can sustain a vacuum level of $1E-4$ Torr. Therefore, we would like to highlight that achieving an entirely oxygen-free environment poses significant challenges, whether within a glove box or under a $1E-7$ Torr vacuum.

In line with our experimental approach aimed at mimicking industrial processes (which is not supposed to involve glove boxes), we stored samples in the deposition queue within a load lock operating under a mild vacuum range of $1E-3$ to $1E-4$ Torr.

Before initiating the load lock transfer, it is important to note that the samples had never been exposed to ambient air. Instead, they were transferred through a glove box with an oxygen level of less than 10 ppm. Once located in the load lock, the system undergoes a pumping process to attain the vacuum level of $1E-4$. Subsequently, a gate valve facilitates the transfer of samples into the deposition chamber. During this exchange, the temperature of the C_{60} crucible is around $240^{\circ}C$ (which we assume is a similar condition to the industrial process).

In our system, the practicality of cooling down the source to room temperature without opening the gate valve between the deposition chamber and load lock is compromised, as the system automatically removes the sample from the chamber immediately after

deposition, without allowing time for cooling. Nevertheless, we anticipate observing a performance drop in the subsequent deposition cycle, attributable to the coalescence effect explained in the manuscript. To answer the reviewer's question, we performed a new experiment to understand how heated/cooled powders affect the performance after the first cycle (following the reviewer's guide on this). Figure R4 shows how the cooling down of the crucible decreases the V_{oc} and FF of the devices. The results confirm our previous observations.

Figure R4: Statistical distributions of V_{oc} and FF for single-junction perovskite solar cells using fresh and cooled thermally cycled as-received C_{60} powder.

5. Although the efficiency of tandem solar cells is very high, the efficiency of single-junction perovskite solar cells is relatively low, especially the open circuit voltage, even after introduce CaF_2 as interlayer in figure s11. How can such inefficient single-junction devices achieve efficient tandem devices?

We believe there might be a misunderstanding here. The perovskite layers showed in Figures 1b and Figure S11 (now has been updated as S13) for the single-junction devices were fabricated using a hybrid method involving evaporation and spin coating. This approach differs from the one-step spin coating method that serves as our baseline for tandem cells, which is also used for high PCE cells (we note here the focus of this work is not perovskite itself).

In the hybrid method, the inorganic scaffold $PbI_2/CsBr$ layer is co-evaporated onto the substrate and subsequently transformed into the perovskite layer by introducing an ethanolic solution containing $MACl$, FAI , and $FABr$. This detail was explicitly highlighted in the caption of Figure 1, which reads: "Here, the perovskite layers of single-junction PSCs were fabricated using a hybrid method that consists of a two-step process. Initially, an inorganic template was evaporated, and then a solution conversion step was employed to accomplish conversion into the perovskite phase." We intentionally employed this hybrid method due to its ability to yield more homogeneous and reproducible devices on glass substrates (despite its lower performance than the one-step solution method), resulting in narrower device statistics. This choice played a critical role in our ability to conclude the repeatability issue observed during multiple C_{60} depositions.

6. In figure 1d, the HTL is NiO/MeO-2PACz. However, in Perovskite/silicon tandem solar cells fabrication, the HTL is 2PACz. Is there any difference for these two hole-selective materials?

The perovskite layer of single-junction devices in Figure 1d (main text) is deposited by evaporation/spin coating (hybrid method). During our optimizations, we found the suitable HTL combination for this method is NiO_x/MeO-2PACz among many other options (we have not included this data here as it is out of scope).

However, the optimal HTL for one-step solution-processed perovskite/silicon tandem solar cells is 2PACz. The perovskite layer in perovskite/silicon tandem solar cell is deposited by the spin coating method. We note that our recently published article which delivers 32.5% tandem cells is based on a one-step solution-processed perovskite is also utilizes 2PACz as an HTL (doi.org/10.1038/s41586-023-06667-4). We believe this is well demonstrated in our previous works.

Reviewer #3

This paper aims to elucidate the effects of repeated C₆₀ evaporation on single-junction and silicon/perovskite tandem solar cell devices, with a particular focus on the high efficiency (31%) achieved in the silicon/perovskite tandem configuration. The study appears to have meticulously conducted film analyses and calculations concerning the reuse of C₆₀, as depicted in Fig. 2 and Fig. 3. If the conclusions of this study prove valid, they could offer invaluable insights to numerous research groups employing C₆₀ and industrial institutions endeavoring to commercialize perovskite single-junction or tandem devices with a p-i-n configuration. However, a substantial set of questions and concerns arise concerning the primary aspect of this paper - the device data intended to substantiate the main claims of the authors. Addressing these concerns is imperative for mandatory revision, as outlined below, before I could recommend the publication of this manuscript in Nature Communications.

We would like to thank to the reviewer for the constructive comments and suggestions to further improve the quality of our manuscript.

Firstly, the pivotal data that supports the author's conclusions, as presented in Fig. 1, appears insufficient to robustly validate their claims. If I understand correctly, for single-junction data, the authors fabricated the half-device (ITO/HTL/Perovskite) within the same batch. Subsequently, they evaporated C₆₀ for the first set of substrates, stored them in a load lock (under low-vacuum conditions), evaporated C₆₀ for the second set of substrates from the same batch as the first C₆₀ devices set, similarly stored them in a load lock, and repeated this process up to the eighth C₆₀ set before finally introducing BCP/Ag and conducting the experiment. This approach seems appropriate for assessing the influence of the number of C₆₀ depositions. However, at this juncture, I'm curious that how many times the authors repeated this experiment, and the specifics of the experimental design. To ensure that the repeated use of C₆₀ genuinely affects device performance, it is crucial for the authors to replicate this process in separate experiments at least 3-4 times (not for 3-4 substrates at one time) using the same batches of NiO_x/SAM/Perovskite layers. Explicitly detailing the steps taken to ensure controlled conditions that exclusively account for the impact of C₆₀ reuse, while excluding other potential performance variations, is essential to convince the readers.

We do agree with the reviewer that the repeatability of our observations is quite critical to draw a conclusion, which we were also quite careful before elaborating our interpretations. Before starting a more detailed interpretation, we repeated this experiment 3 times. First time, with 1.68 eV hybrid process single-junction perovskite solar cells as shown in Figure R5a. To verify the universality of the C_{60} behavior after multiple deposition cycles, we repeated the same experiment with 1.68 eV and 1.55 eV one step solution-process perovskite solar cells as shown in Figure R5b and c, respectively. Furthermore, we carried out the same experiment with solution-processed perovskite/tandem solar cells (Supplementary Information Figure S3).

However, to build a further confidence on our work, following the reviewer's recommendation, we performed the experiment one more time as shown in Figure R5d. Figures R5 b, c and d have been added to the Supplementary Information as Figure S2. So, we believe that the readers will have full confidence about the repeatability/consistency/universality of this issue. We thank the reviewer once more for giving us an opportunity to bring a further confidence to our results.

Figure R5: Statistical distributions of V_{oc} and FF for, a) single-junction hybrid 1.68 eV perovskite solar cells, b) 1.68 eV solution-processed $Cs_{0.05}FA_{0.8}MA_{0.15}Pb(I_{0.745}Br_{0.255})_3$ perovskite solar cells c) 1.55 eV solution-processed $Cs_{0.03}(FA_{0.90}MA_{0.10})_{0.97}PbI_3$ perovskite solar cells, d) 1.68 eV single-junction hybrid perovskite solar cells.

Additionally, measures to mitigate variables stemming from the time spent in the load lock need to be established (e.g., comparing substrates stored without using the load lock and substrates stored in the load lock for the same duration as the deposition times)

for the 1st to 8th C₆₀ experiments, followed by experiments using fresh C60 to verify consistency).

We thank the reviewer for raising this interesting aspect. To answer this question, we designed a device fabrication batch by keeping the samples in;

- 1) Glove box with O₂ and H₂O levels <0.1 ppm
- 2) Load lock by keeping the samples in vacuum level 1E-4 Torr

for 5 hours (assuming an equivalent time for one lot). Later, we deposited C₆₀ layers by using as-received fresh powders in the same batch. Figure R6 summarizes the V_{oc} and FF of those devices.

Figure R6: Influence of the perovskite absorber storage on cells performance: Statistical distributions of V_{oc} and FF for fresh substrates and vacuum- and N₂ glove box-stored perovskite absorbers before C₆₀ deposition.

We found that the V_{oc} and FF of vacuum-stored substrates were unaffected by the vacuum as shown in Figure R6. The same results were obtained when we compared the performance of the fresh perovskite substrates and N₂-stored perovskite substrates. (Figure R6 has been added to the Supplementary Information as Figure S16 a and b).

These findings confirm that the origin of V_{oc} and FF losses is not due to the storage, it is related to the changes in C_{60} powder after multiple deposition cycles.

Furthermore, a major factor contributing to the lack of persuasiveness in the device data is the behavior observed in the Fig. 1 graph. A drop is evident between the first and second data points, followed by almost consistent values between the 2nd and 8th data points, with a subtle decrease at the end. This pattern does not match well with the QFLS data in Figure 2. Efficiency remains nearly constant between the 2nd and 7th data points (excluding the 1st and 8th). It is imperative to elucidate whether the divergent efficiencies at the 1st and 8th data points are coincidental or genuinely attributed to the reuse of C_{60} . The tandem data in Fig. 1 does not exhibit this discrepancy as prominently. The J-V curves in the tandem data are almost superimposed, making it challenging for readers to accept the efficiency drop posited by the authors. While a numerical efficiency drop of 0.6% presented, it's a small value; it might be more compelling to represent this reduction through the distribution illustrated in the supplementary materials.

Regarding Figure 1c, it is noteworthy that the mean average open-circuit voltage values resulted in a slight drop from the 2nd to the 4th deposition. Subsequently, from the 5th to the 8th deposition, a significant decline was observed, which aligns with the results of the QFLS (Quasi Fermi Level Splitting) measurements. In our QFLS analysis, we specifically compared the results obtained from the 1st, 6th, 7th, and 8th depositions to reduce the experimental load.

Furthermore, the PCEs exhibited a slight decrease due to the reuse of C_{60} , primarily affecting the open-circuit voltage, as it did not significantly impact the short-circuit current densities. To validate our findings, we conducted the experiments four times.

1. We reported the repeated deposition results in Figure R5.
2. To further verify our results independent from the perovskite absorber layers, we repeated these experiments with solution-processed perovskite absorbers with energy bandgaps of 1.55 and 1.68 eV, and again we observed the same trend (see results in S2)
3. Later, we considered adding a CaF_2 layer between the perovskite and C_{60} layers thinking this contact displacement can solve the performance loss issues (which we understood later the C_{60} itself has degenerated after repeated processes). Even in this case, we saw the same trend as shown in Figure S13.
4. Following the reviewer's suggestions, we repeated this experiment one more time with 1.68 eV hybrid perovskite absorber and gave the results, Figure S2.

Concerning the tandem devices, the J-V curves represent the best-performing device with fresh and thermally aged as-received C_{60} .

Notably, we observed the same behavior in tandem devices as in the single-junction devices, as illustrated in Figure S3. So, we believe the achieved results show a strong consistency.

I acknowledge that repeating the entire experiment for all conditions is a considerable undertaking. Therefore, I recommend focusing on replicating 2-3 types of devices: 1) using fresh C_{60} , and 2) applying 8 cycles of heating and cooling for C_{60} (w/o fabricating device), followed by fabricating the device repeatedly using the same batch of perovskite layers, while verifying the absence of any load lock storage influence.

Additionally, it could be beneficial to introduce one more condition in the middle, involving 4 cycles of heating and cooling, if feasible for the authors.

We thank the reviewer for helping us to further prove the repeatability of our observations. We performed 8 cycles of the process without fabricating devices and used this powder to fabricate a new deposition. For this, we used identical perovskite half-device stacks fabricated in the same batch. It seems that even in this case, the V_{oc} and FF of fabricated perovskite solar cells follow the same trend as shown in Figure R7. Our experiments performed for the other queries of the reviewer (e.g., storing in vacuum and N_2 , repeating, etc) are complementary to this answer.

Figure R7: Statistical distributions of V_{oc} and FF of fabricated solar cell devices with fresh C_{60} powder and cooled C_{60} powder after 8 thermal cycles.

This type of data could be statistically integrated into Figure 1. If the authors' claims hold true and similar results are achieved, the disparities could be more explicitly displayed on the Fig. 1 graph, featuring an ample number of data points. Moreover, with the careful specification of the controlled conditions mentioned earlier, readers are likely to find greater resonance with the authors' assertions alongside the improved trends demonstrated by the enhanced Fig. 1 graph.

We express our gratitude to the reviewer for their valuable recommendation. After thorough deliberation with several authors, we have opted to include this information in the Supplementary Information. This choice was made to prevent excessive space consumption within the main text and to maintain the readers' focus on the in-depth interpretations. We trust that the reviewer will appreciate our decision.

REVIEWERS' COMMENTS

Reviewer #1 (Remarks to the Author):

The revised manuscript addresses the three reviewers' comments, which reinforce the significance of the research findings and contribute to a comprehensive understanding of the impact of C60 aggregation on perovskite solar cell performance. The inclusion of the updated data and figures significantly enhances the manuscript's quality and provides a comprehensive understanding of the influence of C60 aggregation on perovskite solar cells. The authors addressed my queries satisfactorily and recommended for publication.

Reviewer #2 (Remarks to the Author):

During this revision, the authors have well addressed all my concerns. I have no further comment and recommend accepting this paper.

Reviewer #3 (Remarks to the Author):

I applaud the authors for their thorough and conscientious responses to the reviewers' comments. It is evident that the authors have effectively addressed my previous remarks, primarily through repeated experiments, which have solidified their claims as factual. Therefore, I strongly recommend that the manuscript be published in Nature Communications at its current stage. The additional comments provided below are my personal suggestions aimed at further enhancing the clarity of the authors' arguments. No further response is necessary.

With regard to Figure 1e, my intention was to point out the potential confusion for readers in interpreting the JV curve in this figure. Given that the JV curves appear almost identical, a casual observer might erroneously conclude that there is a difference in the single-junction but not in the tandem configuration when C60 is reused. To mitigate this potential misconception, it might be advisable to exclude the JV curve and present the statistical data in Figure S3. Alternatively, even if the tandem data is removed from Figure 1, it may help streamline the core argument.

Regarding the last comment, the suggestion is not to introduce the retested data (Fig. R5, R7) as a separate figures. Instead, I propose integrating these data points into Figure 1c, as it could yield a more statistically robust outcome. I believe that this approach would better emphasize the degradation in efficiency resulting from the reuse of C60.

REVIEWER COMMENTS

Reviewer #1

The revised manuscript addresses the three reviewers' comments, which reinforce the significance of the research findings and contribute to a comprehensive understanding of the impact of C60 aggregation on perovskite solar cell performance. The inclusion of the updated data and figures significantly enhances the manuscript's quality and provides a comprehensive understanding of the influence of C60 aggregation on perovskite solar cells. The authors addressed my queries satisfactorily and recommended for publication.

We thank the reviewer for reviewing the manuscript and appreciate the recommendation for the publication.

Reviewer #2

During this revision, the authors have well addressed all my concerns. I have no further comment and recommend accepting this paper.

We are grateful for the reviewer's comment and recommendation for publication.

Reviewer #3

I applaud the authors for their thorough and conscientious responses to the reviewers' comments. It is evident that the authors have effectively addressed my previous remarks, primarily through repeated experiments, which have solidified their claims as factual. Therefore, I strongly recommend that the manuscript be published in Nature Communications at its current stage. The additional comments provided below are my personal suggestions aimed at further enhancing the clarity of the authors' arguments. No further response is necessary.

We thank the reviewer for the feedback which helps to further improve the manuscript quality. We appreciate it.

With regard to Figure 1e, my intention was to point out the potential confusion for readers in interpreting the JV curve in this figure. Given that the JV curves appear almost identical, a casual observer might erroneously conclude that there is a difference in the single-junction but not in the tandem configuration when C₆₀ is reused. To mitigate this potential misconception, it might be advisable to exclude the JV curve and present the statistical data in Figure S3. Alternatively, even if the tandem data is removed from Figure 1, it may help streamline the core argument.

Indeed, we provide Figure 1e with the table of photovoltaic parameters which clearly shows the 1 % drop in FF % from as-received fresh C₆₀ to thermally cycled as-received C₆₀. In addition, the two J-V curves of fresh and thermally cycled as-received C₆₀ don't overlap and aren't identical,

especially for the open circuit voltage point and maximum power point. Therefore, we prefer to keep these J-V curves still in the main text. We hope that the reviewer will understand our reasoning for this.

Regarding the last comment, the suggestion is not to introduce the retested data (Fig. R5, R7) as a separate figures. Instead, I propose integrating these data points into Figure 1c, as it could yield a more statistically robust outcome. I believe that this approach would better emphasize the degradation in efficiency resulting from the reuse of C₆₀.

We thank the reviewer for the suggestion to arrange the figures. We note that the statistical data in Figure 1c showed the same degradation trend as in R5 and R6. In Figure 1c, exclusive to the main text, we provided all device parameters including J_{sc} and PCE results. In that figure, we wanted to show the effect for both single-junction and tandem cells. Therefore, we shared the J-V curves of the two representative devices with device statistics in Figure 1. Regarding the repeated measurements for various samples, we gave only the critical parameters (FF and Voc) in supplementary information so as not to inflate the number of panels.